

# Grab what you can—an evaluation of spatial replication to decrease heterogeneity in sediment eDNA metabarcoding

Jon T. Hestetun[1], Anders Lanzén[2,3] and Thomas G. Dahlgren[1,4]

[1] NORCE Environment, Bergen, Vestland, Norway
[2] Marine Ecosystems Functioning, AZTI, Pasaia, Basque Country, Spain
[3] IKERBASQUE, Basque Foundation of Science, Bilbao, Basque Country, Spain
[4] Department of Marine Sciences, University of Gothenburg, Gothenburg, Sweden

Corresponding author
Jon T. Hestetun,
jhes@norceresearch.no

## ABSTRACT

Environmental DNA methods such as metabarcoding have been suggested as possible alternatives or complements to the current practice of morphology-based diversity assessment for characterizing benthic communities in marine sediment. However, the source volume used in sediment eDNA studies is several magnitudes lower than that used in morphological identification. Here, we used data from a North Sea benthic sampling station to investigate to what extent metabarcoding data is affected by sampling bias and spatial heterogeneity. Using three grab parallels, we sampled five separate sediment samples from each grab. We then made five DNA extraction replicates from each sediment sample. Each extract was amplified targeting both the 18S SSU rRNA V1–V2 region for total eukaryotic composition, and the cytochrome c oxidase subunit I (COI) gene for metazoans only. In both datasets, extract replicates from the same sediment sample were significantly more similar than different samples from the same grab. Further, samples from different grabs were less similar than those from the same grab for 18S. Interestingly, this was not true for COI metabarcoding, where the differences within the same grab were similar to the differences between grabs. We also investigated how much of the total identified richness could be covered by extract replicates, individual sediment samples and all sediment samples from a single grab, as well as the variability of Shannon diversity and, for COI, macrofaunal biotic indices indicating environmental status. These results were largely consistent with the beta diversity findings, and show that total eukaryotic diversity can be well represented using 18S metabarcoding with a manageable number of biological replicates. Based on these results, we strongly recommend the combination of different parts of the surface of single grabs for eDNA extraction as well as several grab replicates, or alternatively box cores or similar. This will dilute the effects of dominating species and increase the coverage of alpha diversity. COI-based metabarcoding consistency was found to be lower compared to 18S, but COI macrofauna-based indices were more consistent than direct COI alpha diversity measures.

## INTRODUCTION

Ecological analyses of community structure in soft bottom marine benthos are commonly done by sieving grab or box corer samples. Using this method, specimens are manually separated from remaining sediment at a given mesh size fraction, creating a species list based on taxonomic identification of each recovered organism (*ISO, 2014*). However, this approach is relatively time consuming. It also relies on access to specialized taxonomic expertise and good local species descriptions, and omits ecologically important meiofauna and microorganisms from the analysis (*Schander & Willassen, 2005*). For these reasons, methods based on environmental DNA (eDNA), such as metabarcoding, have long been considered as an attractive alternative for characterization of benthic communities (e.g., *Aylagas et al., 2016*; *Baird & Hajibabaei, 2012*; *Di Battista et al., 2020*; *Faria et al., 2018*; *Lanzén et al., 2016*).

While metabarcoding has many potential advantages, the methodology employed at all steps in sampling, processing and analysis greatly influences study results (*Cristescu & Hebert, 2018*; *Kelly, Shelton & Gallego, 2019*; *Zinger et al., 2019*), and though a number of studies have investigated the effects of e.g., sampling design (*Penton et al., 2016*; *Zhou et al., 2011*), extraction protocol (*Brannock & Halanych, 2015*; *Djurhuus et al., 2017*; *Lekang, Thompson & Troedsson, 2015*; *Pearman et al., 2020*), primer choice (*Alberdi et al., 2018*; *Andújar et al., 2018*; *Leasi et al., 2018*; *Leray & Knowlton, 2017*; *Piñol, Senar & Symondson, 2019*; *Tang et al., 2012*; *Weigand & Macher, 2018*), reference barcode database coverage (*Hestetun et al., 2020*) and processing (*Smith & Peay, 2014*), sizable knowledge gaps remain. Robust sampling designs are required to win acceptance for the method from management bodies (*Roussel et al., 2015*) and are also a prerequisite for reproducibility, which was recently found to be low in many terrestrial and freshwater eDNA barcoding studies (*Dickie et al., 2018*).

For eDNA that is extracted directly from sediments, an area of specific concern is the amount of source material that is necessary for adequate community characterization. When grab or core samples are processed for morphological analysis, a relatively large volume of sediment is typically sieved at a certain mesh size (commonly 0.25 to one mm). A van Veen grab recovers around 15–20 liter of sediment and a USNEL box corer more than 100 liters. Thus, morphological community analysis is based on the captured community from a comparatively large sample, even when considering that most of the captured organisms are found closer to the sediment surface. On the other hand, eDNA extraction is limited to subsampling a total source volume that is several magnitudes lower, typically measured in grams. At this scale, the heterogeneous distribution of extracellular DNA, mixed with smaller organisms in the source sediment, introduces a strong random sampling effect. This raises the question of to which degree the source material used for eDNA studies can adequately represent the biological community captured by the total grab

or core volume, not to mention the sampling locality itself (*Lanzén et al., 2017*; *Nascimento et al., 2018*).

In the case of coastal or shelf depth morphological sampling, 2–5 van Veen grab replicates are often used to account for spatial heterogeneity of macrofauna at meter-scale (*Hatlen et al., 2019*; *NS, 2009*). For eDNA sampling, different levels of spatial heterogeneity can be expected from smaller scales within a grab (millimeters to centimeters), to the intermediate meter scale of separate grab parallels. Furthermore, the relative extent of this variability can be expected to be different in microorganisms compared to metazoans. DNA extraction replicates are typically used to account for heterogeneous distribution, and a handful of studies have investigated the incremental increase in observed diversity with additional extraction replicates or sediment volume (*Brannock & Halanych, 2015*; *Lanzén et al., 2017*; *Nascimento et al., 2018*). Less attention has been given to investigating the relative effect on data variability at different levels of spatial distance, such as within and between replicate grab samples from the same station.

As indicated previously by e.g., *Lanzén et al. (2017)*, the effect of eDNA patchiness is likely more pronounced for meio- and macrofauna compared to single celled organisms. Hence, for metazoan metabarcoding studies using e.g., the COI marker, a common strategy is to recover the animals from the sediment prior to DNA extraction (e.g., *Atherton & Jondelius, 2020*; *Elbrecht & Leese, 2015*; *Faria et al., 2018*; *Fonseca et al., 2010*; *He, Sutherland & Abbott, 2021*; *Lejzerowicz et al., 2015*). Conversely, when extracting eDNA directly from sediment, it is more common to use molecular markers that target microorganisms, such as the eukaryotic 18S rRNA or the prokaryotic 16S rRNA genes (e.g., *Guardiola et al., 2016*; *Keeley, Wood & Pochon, 2018*; *Lanzén et al., 2016*; *Lanzén et al., 2020*; *Sinniger et al., 2016*; *Stoeck et al., 2018*). In several recent sediment extract studies using multiple markers, however, ribosomal markers have been supplemented with e.g., COI with promising results (*Atienza et al., 2020*; *Laroche et al., 2020*; *Macher et al., 2018*; *Mauffrey & Cordier, 2020*). These studies suggest that sediment extracts may provide useful results despite a potential increase in spatial heterogeneity, which is significant as direct sediment extraction is a faster approach than separating organisms from the sediment, and thus especially suitable when processing a large number of samples for e.g., marine monitoring purposes.

Here, we conducted a comparative experiment with samples from three grab parallels at one North Sea sediment monitoring station located in the general vicinity of an offshore oil drilling platform. The sampling design was chosen from a practical perspective, including both sediment volume and spatial distance as parameters of interest: Metabarcoding sampling and data analysis was performed at three spatial scales: (A) comparison of single DNA extraction replicates from one sediment sample from each grab (extract reproducibility), (B) comparison of separate sediment samples from different parts of the same grab using pooled extraction replicates (sediment sample reproducibility), and (C) comparison of all data from each grab (grab level reproducibility) (Fig. 1). Metabarcoding was done separately using total eukaryotic SSU 18S rRNA V1–V2 region and metazoan cytochrome oxidase subunit I (COI) markers, to investigate the effects of spatial distance for the total eukaryotic community and for metazoans only. This sampling design was chosen in order to evaluate to which degree metabarcoding data from single and pooled
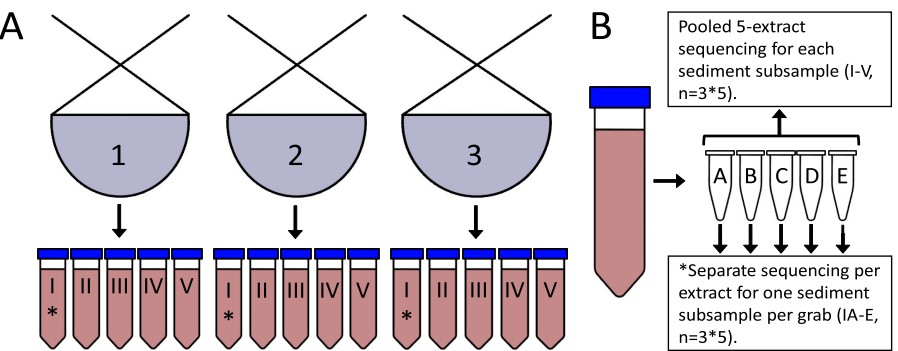

**Figure 1 Study sampling design.** (A) Five separate 50 g sediment samples each were taken from three grab parallels at the offshore monitoring station GK-DA-01. (B) Each sediment sample ($n = 15$) was stirred by hand before subsampling five 0.5 g sediment PowerSoil DNA extracts ($n = 75$). DNA extracts from one sediment sample per grab (marked with an asterisk) were sequenced separately to investigate intra-sediment sample heterogeneity ($n = 3 * 5$ per marker). To examine heterogeneity within and between grabs, the five DNA extracts from each sediment sample were pooled prior to PCR and sequencing ($n = 3 * 5$ per marker).

DNA extracts were representative both of a particular sediment sample ("sediment sample level"), the grab of origin ("grab level"), and a the total multi-grab community dataset ("station level"). Results were evaluated using common alpha and beta diversity measures (richness, Shannon diversity, and Bray—Curtis pairwise dissimilarity). For the COI data, we also included existing macrofaunal biotic quality indices in the analysis.

The main aim of this study was to get a clearer picture of how eDNA sampling at different spatial scales impact perceived variation of organism distribution in metabarcoding data for marine sediment benthos at shelf depth, both for total eukaryotes and metazoans, to inform sampling design and intensity for further sediment eDNA studies in similar habitats.

## MATERIALS & METHODS

### Field sampling

Sediment samples were collected using a 0.1 m$^2$ van Veen grab during the 2018 environmental region II offshore monitoring campaign on the Norwegian Shelf, from three grab parallels at the monitoring station GK-DA-01 (58.5757°N, 1.6973°E, 116 m depth), here designated G1–G3. For each grab, five separate 50 g sediment samples (15 total) were collected from the top two cm of sediment, immediately frozen and kept at −20 °C until DNA extraction, here designated with Roman numerals I–V (Fig. 1A).

This station is part of the routine Norwegian offshore monitoring programme (*Norwegian Environment Agency, 2020*), and physiochemical parameter measurements and morphological species identification were also performed as part of the regular monitoring, which detected no faunal impact disturbance at this station (*Hatlen et al., 2019*). The sediment was classified as fine sand ($\varphi = 3.03$, silt and clay = 11.14%, sand = 83.13%, gravel = 5.73%, TOC = 0.48%).

## Lab processing

Sediment samples were thawed at +4 °C and stirred by hand for approximately one minute prior to DNA extraction. Five 0.5 g sediment replicates, here designated with the letters A-E, were collected from each sediment sample (75 total) and extracted using a semi-automated protocol with Qiagen PowerSoil tubes and C1 solution, a Precellys homogenizer (3 × 6000 rpm for 40 s). After centrifugation (10K rpm for 1 min) we used a Qiagen QIAsymphony SP robot (DSP DNA kit, Tissue LC protocol) for remaining extraction steps. Extract concentrations were measured using a Qubit 3.0 fluorometer (Thermo Fisher Scientific). Amplification and sequencing were performed on each individual extract from one sediment sample per grab (n = 3*5) and by pooling all five extracts from each sediment sample (n = 3*5) for a total of 30 sequencing results per marker (Fig. 1B). Average percent deviations of extract concentrations for each sediment sample were minor (2.65−6.17%). Hence, we used equal volumes of each extract in extract pools.

We used two markers for PCR amplification: the 18S rRNA gene V1–V2 region (∼350–400 bp) with primers SSU_F04mod (5′-GCTTGWCTCAAAGATTAAGCC-3′) (Cordier pers. comm.) and SSU_R22 (5′-CCTGCTGCCTTCCTTRGA-3′) (*Sinniger et al., 2016*), and the cytochrome oxidase subunit I (313 bp) with primers mlCOIintF-XT (5′-GGWACWRGWTGRACWITITAYCCYCC-3′) (*Wangensteen et al., 2018*) and jgHCO2198 (5′-TAIACYTCIGGRTGICCRAARAAYCA-3′) (*Geller et al., 2013*). PCR was performed using the KAPA3G Plant PCR kit (Kapa Biosystems) with adapter-linked primers including 12 random bases in order to improve variance during the first sequencing cycles. Annealing temperatures for 18S and COI were 57 °C and 45 °C respectively. COI primer concentration was tripled relative to 18S to account for the high number of primer ambiguous bases. Illumina dual index TruSeq i5/i7 barcodes were used for library preparation with equimolar PCR product concentration, and extraction and PCR negative controls were used to detect contamination during processing. Sequencing was performed on an Illumina MiSeq instrument using v3 with 300 bp chemistry at the Norwegian Sequencing Centre (University of Oslo, Norway). Demultiplexing was done by the sequencing center. Raw sequence data were uploaded to the NCBI Sequence Read Archive (SRA) as BioProject ID PRJNA704795, BioSample IDs SAMN18055833–41 and accession numbers SRR13781971–SRR13782030.

## Data processing and analysis

Raw FASTQ files were quality checked using FastQC v0.11.8 (*Andrews, 2010*), with further processing using a slightly modified version of the protocol from *Lanzén et al. (2020)*. Briefly, *vsearch* v2.111.1 (*Rognes et al., 2016*) was used for read pair merging, filtering, singleton removal and chimera detection with maximum 20 and 40 mismatches allowed for 18S and COI respectively. Primers were then trimmed with Cutadapt v1.18 (*Martin, 2011*) and reads with incomplete primer sequences or those outside the length range 330–450 bp (18S) or 274–330 bp (COI) were discarded. Clustering was done using SWARM v2.2.1 (*Mahé et al., 2015*) with default settings i.e., a maximum linkage difference (-d) of 1 and 3 for the fastidious step (-b), and post-clustering curation using LULU (*Frøslev et al., 2017*). Taxonomy was assigned using CREST (*Lanzén et al., 2012*) with the SilvaMod v128 database for 18S and the

BOLD (*Ratnasingham & Hebert, 2007*) database for COI, accessed February 2018 and adapted to CREST as part of release 3.2.1 (https://github.com/lanzen/CREST). Workflow scripts are available in the GitHub repository https://github.com/lanzen/Metamon1/ (vsearch_prep_18S_V1V2.sh, vsearch_prep_CO1XT.sh, SWARM_and_LULU_SSU.sh and SWARM_and_LULU_CO1.sh).

Potential contaminant OTUs were identified and removed based on both abundance profiles in the PCR and extraction blanks, using decontam (*Davis et al., 2018*), removing one 18S OTU but none from COI. Additional filtering in R included reducing cross-contamination by removing OTU occurrences at very low (<1%) abundance compared to the average of all samples, analogous to the UNCROSS algorithm (*Edgar, 2016*). Further, OTUs below the minimum classification threshold or belonging to probable contaminant taxa (Insecta, Mammalia, Myxini, Arachnida, Actinopterygii, Collembola and Onychophorida) were removed. As we wanted to examine metazoan spatial distribution specifically, COI OTUs not classified to metazoan phylum rank or lower were removed. Finally, prior to alpha and beta diversity analysis, OTUs with an average relative abundance across samples corresponding to less than 3 copies in the sample with lowest sequencing depth (0.003% for 18S; 0.1% for COI) were also removed.

## Statistics

Bray–Curtis pairwise dissimilarity values were calculated based on relative OTU abundances. Non-metric dimensional scaling (NMDS) was carried out using the R package vegan v3.2.1 (*Oksanen et al., 2019*; function metaMDS), as well as species accumulation curves. Pairwise dissimilarities were compared group-wise (between extracts, pooled intra-sample replicates and grab) using Wilcoxon rank sum tests.

For direct comparison with morphological values from the standard monitoring surveys in the area (*Hatlen et al., 2019*), Shannon diversity estimates were calculated using the base 2 rather than the natural logarithm. Shannon diversity values were calculated for both 18S and COI data using the R BBI package (*Cordier & Pawlowski, 2018*). For COI, BBI was also used to calculate several macrofaunal biotic indices based on taxonomic classification of the macrofauna reads in the metazoan data, including the AZTI Marine Biotic Index (AMBI) (*Borja, Franco & Pérez, 2000*), and the local Norwegian biotic indices NQI1 (*Rygg, 2006*), ISI2012, and NSI2012 (*Rygg & Norling, 2013*). In order to compare the taxonomic coverage of the study 18S and COI data with the *Hatlen et al. (2019)* morphological dataset from the same station, morphological species data were downloaded from the MOD database (*DNV GL, 2021*). The Euler diagrams were made using the R eulerr package (*Larsson, 2020*). The R scripts used for all analyses are available in the repository https://github.com/lanzen/Metamon1 (*WP2/R/Intra_plate_heterogeneity_18S.R* and *WP2/R/Intra_plate_heterogeneity_COI.R*).

# RESULTS

## Data overview

In total, 30 samples were sequenced for each marker, sequentially identified here by grab number (ten samples each for G1–G3), sediment sample (I–V, comprising five pooled

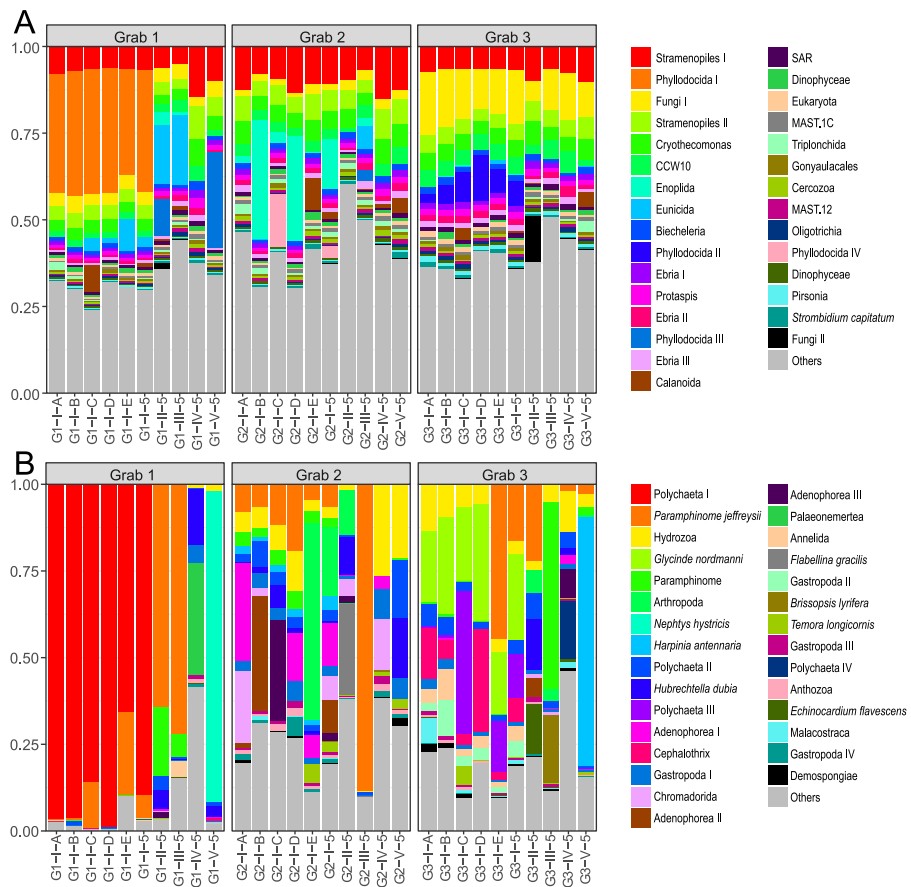

**Figure 2** **Dataset read abundance.** Read abundance and the most abundant OTUs for (A) all 18S metabarcoding data and (B) COI phylum level or lower metabarcoding data at sediment sample (I–V) and individual extract (A–E) level for the entire three-grab dataset.

extracts each) and finally individual extracts from sediment sample I from each grab (A–E). (Table S1).

The total 18S dataset comprised 4,526,427 reads of which 3,778,901 (84%) remained after filtering representing 893,214 unique paired sequences, with 103,321–153,079 filtered reads remaining for individual samples (mean = 125,963; $\sigma$ = 15,287). After swarm clustering 6,371 unique OTUs remained (6358 after taxonomic filtering). The three most common 18S OTUs in the total dataset were an unclassified stramenopile, a member of the polychaete order Phyllodocida, and an unidentified fungus. Other common OTUs included a nematode from the order Enoplida and cercozoan taxa belonging to Cryothecomonas, Ebria and the CCW10 clade. Metazoan taxa displayed higher inter-sample variation: One Phyllodocida OTU dominated all sediment sample I data from the first grab, but was not present in any other samples. Other Phyllodocida and nematode OTUs were also intermittently present at moderate abundances in other samples (Fig. 2).

The total COI dataset comprised 4,833,536 raw sequences of which 3,313,941 (69%) remained after filtering. Removing sequences not classified to metazoan phylum level or

lower removed an additional 76% of reads for a final dataset of 788,499 reads representing 683,012 unique paired sequences, with 3,380–111,487 reads for individual samples (mean = 26,283; $\sigma$ = 28,657). After swarm clustering, 19,870 OTUs remained, of which the majority were removed due to being non-specific reads from bacteria or non-metazoan eukaryotes, leaving 1,759. Here, an unclassified polychaete, corresponding to the Order Phyllodocida designation in the 18S dataset, was the most abundant OTU, due to an almost complete dominance of this OTU in sediment sample I in grab 1. Other abundant OTUs included the polychaete species *Paramphinome jeffreysii*, *Nephtys hystricis* and *Glycinde nordmanni*, as well as members of taxa from other phyla such as Gastropoda, the nematode Adenophorea, Hydrozoa, the amphipod *Harpinia antennaria* and a couple of nemertean taxa (Fig. 2).

At the metazoan phylum level, 21 metazoan phyla were found in both the 18S and COI data, 24 in 18S only, and 17 in COI only. One phylum was found in the morphological dataset only. Similar numbers at class level were 21, 24, 17 and 1, and at order level, 16, 55, 45 and 8. This information, together with the relative abundances of the 18S, COI and morphological datasets at phylum, class and order level, is given in Fig. S1.

## Variability between samples

Bray—Curtis 18S pairwise dissimilarities were the lowest between individual extracts from the same sample (median =0.24), significantly higher between pooled extracts from different samples within the same grab (median =0.39, p < 1E-5), and higher still between pooled extracts from different grabs (median =0.43, $p = 0.02$) (Fig. 3A). The NMDS plot showed clustering according to grab identity, though grab 1 appeared more heterogeneous than grabs 2 and 3. In addition to the Phyllodocida-dominated grab 1 outlier, comprising all sediment sample I extracts, different grab 1 sediment samples were also less similar to each other than corresponding samples from grabs 2 and 3 (Fig. 3B), which appeared to be caused mainly by differences in abundances of large metazoans (Fig. 2A) .

Bray–Curtis COI pairwise dissimilarities were substantially higher than their 18S equivalents. As expected, individual extracts from the same sediment sample had the lowest dissimilarities (median =0.49) while those from pooled extracts from different samples in the same grab were significantly higher (median =0.89, p < 1E-12), consistent with 18S data. In contrast to 18S however, pooled extracts from samples from different grabs were not more dissimilar than those from the same grab (median =0.92) (Fig. 3C). Corresponding to the 18S results, the COI NMDS analysis showed that grab 1, sediment sample I datapoints were outliers in the dataset. For the remaining samples, grab level clustering could not be readily discerned (Fig. 3D).

## OTU coverage

For the entire 18S dataset, 1030 OTUs remained after abundance filtering, comprising all three grabs (see Table S1 for a full overview). The number of OTUs for each of the three grabs ranged from 933–959 (Table 1), with 845 OTUs shared between all grabs, 105 OTUs in two grabs, and 80 in a single grab only (Fig. 4A). For the five individual sediment samples of each of the three grabs, represented by five pooled extraction replicates each ($n = 15$),

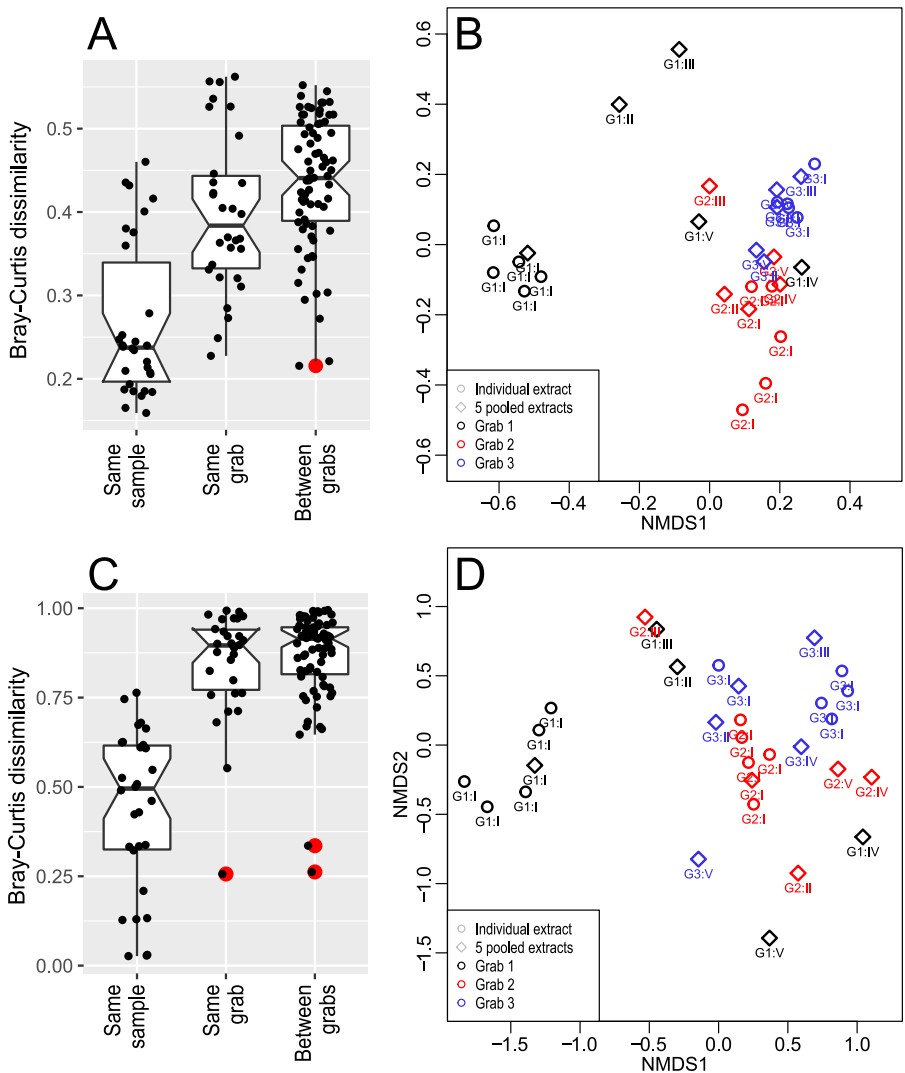

**Figure 3** **Sample beta diversity.** Boxplots of (A) 18S and (C) COI pairwise dissimilarities for individual extracts within one sediment sample from each grab, pooled extracts from each sediment sample from each grab, and between each grab. Dots are individual datapoints, and red dots dataset outliers. NMDS analysis showing relative (B) 18S and (D) COI pairwise distances at individual extract and pooled extract (sediment sample) level.

abundance filtered OTUs ranged from 622–812 (60–79% of total). Single extracts from the three sediment samples were also sequenced separately (one sample from each grab, $n = 15$) and their OTU richness ranged from 641–706 (62–69% of total) (Table 1).

Species accumulation curves for abundance filtered 18S data exhibited an initial steep increase in richness going from one to two samples both when accumulating individual extracts from the same sediment and when accumulating pooled extracts from different sediment samples from the same grab. The slope decrease was slightly less clear at the grab level, indicating higher heterogeneity here compared to individual extracts from the same sediment sample (Fig. 5).

**Table 1 Abundance filtered OTU richness at grab, sediment samples and individual extract level.** Information is given as absolute numbers and percentage of total dataset for the 18S rDNA and COI datasets. Whole grab values are based on sum of pooled extract OTUs, excluding individual same-sediment sample extracts.

| | Single extracts | | | Intra-grab, 5 pooled extracts | | | Whole grab |
|---|---|---|---|---|---|---|---|
| | Min | Max | Median | Min | Max | Median | Value |
| **18S absolute counts** | | | | | | | |
| Grab 1 | 641 | 666 | 659 | 622 | 812 | 752 | 959 |
| Grab 2 | 642 | 656 | 646 | 711 | 782 | 775 | 933 |
| Grab 3 | 666 | 706 | 679 | 695 | 779 | 711 | 933 |
| **18S percentage of total** ($n = 1030$) | | | | | | | |
| Grab 1 | 62% | 65% | 64% | 60% | 79% | 73% | 93% |
| Grab 2 | 62% | 64% | 63% | 69% | 76% | 75% | 91% |
| Grab 3 | 65% | 69% | 66% | 67% | 76% | 69% | 91% |
| **COI absolute counts** | | | | | | | |
| Grab 1 | 35 | 48 | 39 | 37 | 63 | 60 | 83 |
| Grab 2 | 45 | 55 | 49 | 47 | 67 | 48 | 84 |
| Grab 3 | 52 | 59 | 56 | 38 | 59 | 51 | 87 |
| **COI percentage of total** ($n = 108$) | | | | | | | |
| Grab 1 | 32% | 44% | 36% | 34% | 58% | 56% | 77% |
| Grab 2 | 42% | 51% | 45% | 44% | 62% | 44% | 78% |
| Grab 3 | 48% | 55% | 52% | 35% | 55% | 47% | 81% |

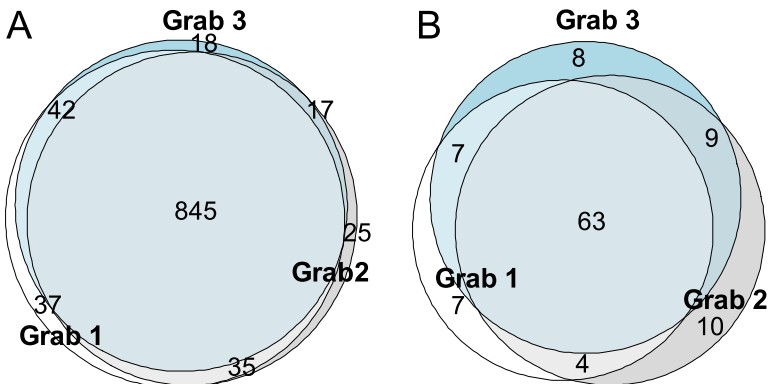

**Figure 4 Grab level OTU coverage.** Euler diagrams of (A) 18S and (B) COI OTUs at grab level, showing the number of unique and shared OTUs for both markers.

For the entire COI dataset at metazoan phylum level or below, 108 OTUs were retained after abundance filtering, comprising all three grabs (see Table S1 for a full overview). The abundance filtered OTU richness per grab was 83–87 (77–81% of total) (Table 1), with 63 OTUs shared between all grabs, 20 OTUs in two grabs, and 25 in a single grab only (Fig. 4B). For individual sediment samples ($n = 15$, made from pools of five extracts each), richness ranged from 37–67 (34–62% of total), while single extract richness from (five from one sub-sample from each grab, $n = 15$) ranged from 35–59 (32–55% of total) (Table 1).
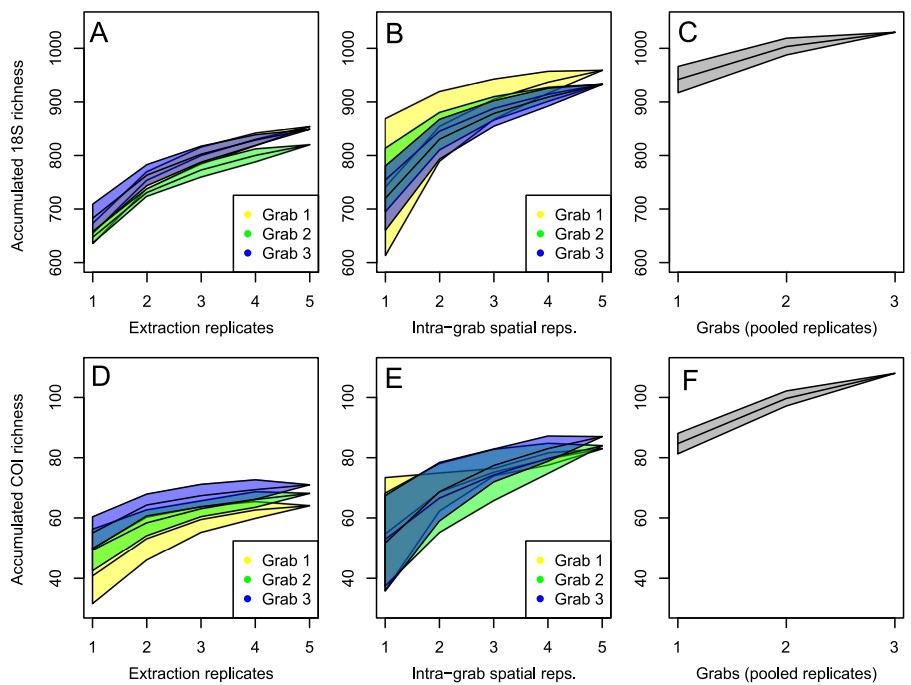

**Figure 5** **OTU aggregation curves.** OTU aggregation curves. Median 18S (A–C) and COI (D–F) richness increase for individual extracts (A, D), and sediment samples per grab (B, E) shown separately for grabs 1–3, and as cumulative total grab data for the whole dataset (C, F).

As seen in the species accumulation curves, COI spatial replicates caused a larger relative increase than in the 18S data, indicating lower sampling coverage for metazoan COI data compared to the eukaryote 18S data (Fig. 5).

### Diversity and sensitivity index values

Shannon values were higher and more consistent for pooled samples relative to single extract samples, and for 18S compared to COI, while COI biotic index values were more consistent between samples than corresponding COI Shannon values. The range of Shannon values for 18S and COI, as well as biotic index values for COI, are given in Table 2. Individual values for each sample can also be found in Table S1 .

# DISCUSSION

18S total eukaryote richness was surprisingly well represented at pooled extract and grab level: Of the total multi-grab 18S dataset, more than 90% of OTUs could be recovered from the sum of pooled samples from each of the single grabs. More practically relevant is that even for the single five pooled extract sediment samples, a median 69–75% of the total abundance filtered 18S OTUs were recovered. Furthermore, even a modest increase in sampling effort substantially increased sampling coverage when combining sediment from different parts of the same grab or different grabs. Thus obtaining representative coverage of the total eukaryote community was feasible with a reasonable amount of sampling effort.

**Table 2  Shannon values for 18S and COI, and biotic index values for COI only.** Data are given as the range and median (in parenthesis) of individual sample values ($n = 5$) for individual extracts, and pooled extracts per sediment sample. Shannon values are given separately for each grab, while biotic index values represent the range in the entire three-grab dataset.

| | 18S Individual extracts | Pooled extracts | COI Individual extracts | Pooled extracts |
|---|---|---|---|---|
| **Shannon** | | | | |
| Grab 1 | 4,86-(5,12)-5,25 | 5,35-(5,76)-6,15 | 0,15-(0,34)-1,31 | 0,68-(1,69)-3,47 |
| Grab 2 | 5,20-(6,01)-6,49 | 6,30-(6,40)-6,86 | 2,89-(3,97)-4,30 | 0,88-(3,98)-4,49 |
| Grab 3 | 6,06-(6,16)-6,30 | 6,14-(6,25)-6,46 | 2,86-(3,19)-4,24 | 1,70-(3,85)-4,66 |
| Total | 4,86-(6,01)-6,49 | 5,35-(6,25)-6,86 | 0,15-(3,17)-4,30 | 0,68-(3,42)-4,66 |
| **Biotic indices** | | | | |
| gAMBI | | | 0,99-(1,54)-2,99 | 0,19-(1,86)-2,98 |
| (pa)gAMBI | | | 0,50-(0,79)-1,15 | 0,47-(0,75)-1,50 |
| ISI2012 | | | 9,14-(9,42)-10,40 | 9,07-(9,30)-10,06 |
| NSI2012 | | | 20,86-(24,23)-28,19 | 20,85-(24,63)-27,41 |
| NQI | | | 0,54-(0,70)-0,75 | 0,58-(0,67)-0,80 |

While the Shannon index and other alpha diversity estimates should not be uncritically used with molecular data, considering that relative abundance can be heavily affected by primer bias and data treatment (e.g., *Krehenwinkel et al., 2017*), it is still useful to analyze the consistency of obtained Shannon diversity values. The values obtained using 18S (median = 6.25) were somewhat higher than those obtained using morphological identification of macroinvertebrates from concurrently collected sieved samples from the same station (median = 5.48; (*Hatlen et al., 2019*). Further, 18S Shannon median values were slightly higher for pooled extracts, representing 2.5 g of sediment, than single 0.5 g extracts. The positive influence of increased sediment volume on sample diversity is well established (e.g., *Nascimento et al., 2018*), and kits such as the Qiagen DNeasy PowerMax Soil Kit or the Soil DNA Isolation Maxi Kit are commonly used to increase sample sediment volume as an alternative to the use of extract replicates as used here and in e.g., *Lanzén et al. (2017)*.

The 18S pairwise sample dissimilarities were lowest in the intra-sediment samples, followed by sediment samples from the same grab, and finally highest for samples from different grabs, with the exception that COI dissimilarities were not significantly higher between grabs than for separate sediment samples within the same grab. Thus we could discern several levels of spatial heterogeneity in our data, meaning that both sampling from different points within a grab and including samples from separate grabs increased community coverage at station level relative to concentrated sampling. Making sample replicates from within the same grab (or alternatively box or multi corer) is not only easier, but our results show that it contributed almost as much to improve sampling coverage as replicating grabs did. However, even for the 18S data, the heterogeneous metazoan distribution in one of our grab replicates shows how entire outlier grabs can substantially and randomly influence perceived sample variability when employing markers that capture metazoans in addition to single-celled eukaryotes. Thus, spatial replication is ideally needed even at grab level. Alternatively, another viable strategy for reducing spatial heterogeneity

is to employ eukaryote markers that minimize the metazoan complement. In their work on Mediterranean deep-sea sediment, *Guardiola et al. (2015)* compared both 18S V7 data between multi corer mini core replicates, and between replicate box corers from the same station. Compared to the same and separate grab replicate mean dissimilarities here of 0.39 and 0.43 using 18S V1–V2 (but both around 0.9 for metazoans for COI), Guardiola et al. reported higher mean Bray-Curtis dissimilarities of around 0.6 and 0.7 respectively, though differences in source habitat, sample design and processing to this study make direct comparison difficult. Finally, several studies have sequenced prokaryote communities (e.g., *Aylagas et al., 2017*; *Lanzén et al., 2020*), though we are not aware of any study where prokaryote heterogeneity in marine sediments is examined at the spatial scapes examined here.

In contrast to 18S, COI metabarcoding data from sediment eDNA typically contains a high number of prokaryote and unassignable eukaryote reads, as observed here and previously by e.g., *Collins et al. (2019)* and *Mauffrey et al. (2020)*. We note that removing non-target reads to focus exclusively on metazoans assigned at phylum rank or lower drastically reduced total read depth to a very variable extent in different samples, leaving 3–67% of total reads for further analysis; a consistent challenge for COI that is difficult to avoid without concentrating metazoan DNA through using community (e.g., bulk) samples rather than sediment. Heterogeneity in the COI dataset was much higher than in corresponding 18S samples, and dissimilarities between separate sediment samples from the same grab were not significantly lower than between grabs. Though around half of total COI OTUs could typically be found in individual sediment samples, coverage was consistently lower than 18S for all levels of sampling. While the reduction in read depth due to non-target reads probably plays a role here, these findings also support the conclusion from e.g., *Lanzén et al. (2017)* that multicellular metazoans have a more heterogeneous distribution than single-celled eukaryotes at the scale examined here.

A common argument for including metazoans in eDNA impact monitoring studies is the ability to use existing macrofauna-based biotic indices such as AMBI, or local variants such as the Norwegian ISI2012, NSI2012 and NQI1 indices. The standard monitoring survey found no disturbed fauna at this station based on alpha diversity, biotic index values and an overall qualitative assessment of the morphological species data (*Hatlen et al., 2019*). Applying the same biotic indices to identifiable macrofauna in our COI data generated results that were mostly consistent with this observation ("no disturbed fauna"): The obtained gAMBI values (median 1.68) would classify the corresponding sample as "unpolluted" (0.0–1.2, dominate by ecogroup I) or as "slightly polluted" (1.2–3.3, dominated by ecogroup II/III) (*Borja, Franco & Pérez, 2000*). Calculated on presence-absence data only, (pa)gAMBI values indicated a slightly better status with less variation across all samples, classifying all but one sample as "unpolluted". Though originally developed for Norwegian coastal sediments, and thus not completely representative of an offshore locality such as the station studied here, the most consistent biotic index values were observed for the indices ISI2012 and NSI2012, where all samples, according to the Norwegian implementation of the Water Directive, corresponded to "very good" or "good" environmental status (*Direktoratsguppen vanndirektivet, 2018*). The compound

NQI index, incorporating AMBI (*Rygg, 2006*), was less consistent (corresponding to "very good" to "moderate"). As a whole, COI-derived biotic index values were comparable to classification of biotic index results from the morphological species data ("good" to "very good") (*Hatlen et al., 2019*).

While there was still some variation between samples, macrofaunal index values were thus more consistent than other COI diversity metrics, possibly because they are calculated from a limited list of taxa with assigned sensitivity status that presumably are represented among the more abundant OTUs in the molecular dataset. Out of the 105 COI OTUs, 71 were represented in at least one biotic index and could be used to calculate quality values, which corresponds well to the relatively good specific reference database coverage of this region (*Hestetun et al., 2020*). Thus our results show that despite the increased noise of the metazoan COI data compared to 18S here, the ability to use existing biotic indices on COI molecular data, as done in previous studies (*Aylagas et al., 2018*; *Lejzerowicz et al., 2015*), can still make sediment COI metabarcoding a useful tool in environmental quality assessment and compliance with current standards.

## CONCLUSIONS

In summary our results show that, given the small amount of source sediment, total eukaryotic diversity was well represented with a manageable number of biological replicates, demonstrating the ability of metabarcoding to consistently detect a eukaryotic core community. However, there was a consistent pattern of increasing heterogeneity with increased spatial distance, and individual extracts and grab sediment samples were vulnerable to large read abundances of single metazoan OTUs. Thus extract and biological replicates from different areas within the same grab and from grab replicates are suggested to both dilute the effect of single dominating OTUs and maximize coverage of the biological community. The inconsistent COI results highlight the challenges in using a metazoan approach for direct sediment extraction, both in terms of non-target sequences and inter-replicate heterogeneity. However, COI biotic index values proved more stable than direct alpha- and beta diversity metrics, adding to the potential value of this marker for environmental monitoring in the context of current monitoring standards.

## ACKNOWLEDGEMENTS

We would like to thank Jessica Louise Ray and Aud Larsen for valuable input on study design and analysis. We would also like to thank Anita Skaarstad and Ane Kjølhamar from Equinor AS and Grethe Kjeilen-Eilertsen and Thomas Merzi from Total E&P Norge AS for project input and support. Finally, we would like to thank Kristin Hatlen and Frøydis Lygre at Stim Miljøtjenester AS for collecting the sediment samples during the 2018 North Sea region II monitoring campaign.

### Funding

The study was supported by the Research Council of Norway (RCN), with additional funding from industry partners Equinor AS and Total E&P Norge AS, through the MetaMon project (RCN no. 280919). The funders had no role in study design, data collection and analysis, decision to publish, or preparation of the manuscript.

### Grant Disclosures

The following grant information was disclosed by the authors:
Research Council of Norway (RCN).
MetaMon project: 280919.

### Competing Interests

The authors declare there are no competing interests.

### Author Contributions

- Jon T. Hestetun conceived and designed the experiments, performed the experiments, analyzed the data, prepared figures and/or tables, authored or reviewed drafts of the paper, and approved the final draft.
- Anders Lanzén conceived and designed the experiments, analyzed the data, prepared figures and/or tables, authored or reviewed drafts of the paper, and approved the final draft.
- Thomas G. Dahlgren conceived and designed the experiments, analyzed the data, authored or reviewed drafts of the paper, and approved the final draft.

### Data Availability

Raw sequence data are available at NCBI Sequence Read Archive (SRA): BioProject ID PRJNA704795, BioSample IDs SAMN18055833–41 and SRR13781971–SRR13782030.

### Supplemental Information

Supplemental information for this article can be found online at http://dx.doi.org/10.7717/peerj.11619#supplemental-information.

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
