# Peer review of "Grab what you can—an evaluation of spatial replication to decrease heterogeneity in sediment eDNA metabarcoding"

_PeerJ, doi:10.7717/peerj.11619_

## Round 0.1 · original submission · Minor Revisions

Dear Jon and co-authors,

I have now received three independent reviews of your study. While all reviewers clearly recognised the quality/novelty of your work, they have collectively raised a number of minor issues that will need to be addressed in your revised manuscript.

Overall, the reviewers have provided you with excellent suggestions on how to improve the manuscript, and I'll be looking forward to receiving your revised manuscript along with a point-by-point response to the reviewers' comments.

With warm regards,
Xavier

·

Basic reporting

No comment

Experimental design

No comment

Validity of the findings

No comment

Additional comments

I have read with interest this article. I found it interesting and worth publishing in PeerJ. It is quite straightforward and the conclusions, particularly the need to subsample within grabs, can be useful for future studies. Although only 30 samples are analysed, the authors compare the outcome of two different markers.
I would have appreciated if the study could have covered a comparison also between different locations, but I understand this is beyond the scope of this work.
I waive anonymity, and include below some comments (none too serious) that can, hopefully, improve the ms
One not-so-trivial comment. It is not clear to me what kind of data is available from Hantlen et al 2019, as the title of this report is in Norwegian, but I understand that this is a study using morphology on the same samples or the same location. If so, and in addition of comparing diversity values and quality indices, can the authors compare the actual results in terms of which species are found in 18S and COI datasets that have been detected morphologically?

Introduction
Line51. This statement needs a qualifying remark … it applies only to soft bottom benthos.
Paragraph in Line 75. It should be mentioned here that for metabarcoding studies there is also the option to sieve or not to sieve the sediment. This point is commented later on, but it would be useful to briefly mention here.
Lines 93-94. Can this ms provide some evidence to confirm or refute the idea that metazoans will have higher heterogeneity that microorganisms? I think so, but couldn’t find it in the Discussion
Line 118 and following. I understand this is not easy to explain, but many similar words are creating confusion. Sentences like “pooled extraction replicates from three spatial whole grab replicates from the same station” are confusing in my view. The terminology is also not consistent with that of the figure. What is a sample? The intra-grab five 50g “units” are referred to as subsamples in text but as samples in figure. What is a sub-sample: the different parts of a grab? Because the same name is used for the 50 g “units” in the text and for the 0.5 g “units” in the figure. Try to explain the terms right at the start and be consistent afterwards.

Methods
If I understand correctly, from the first 50g “unit” of each grab the DNA was extracted and amplified separately for each 0.5g “subunits”, but for the same “unit” the five extracts of the “subunits” have been also pooled and amplified. Can you explain? What volume of extraction product did you use for the separate amplifications and what volume is used for the pooling? Is the pooling performed always with the same volume of extract? Is the amount of DNA in the extracts quantified and volumes adjusted somehow to add the same amount of DNA per “subunit” to the pools?
Why limit the COI analysis to metazoans? This requires some justification
Explain the d parameter used in SWARM… the same for 18S and COI?
Results
Lines 231 and 241. I infer that by “number of sequences” the authors mean the number of reads obtained. It would be interesting to quote also the number of different sequences obtained (i.e., de-replicating reads).
An OTU is a taxon, ideally approaching species-level. Thus, I think it is misleading to say that an OTU is “the polychaete order Phyllodocida” (and similar sentences), because an order is not an OTU. I guess the authors mean “a member of the polychaete order Phyllodocida” (even if it cannot be given a species name). I strongly advice to rephrase this and similar sentences (unidentified Fungi, the nematode order Enoplida…).
Likewise, I also suggest to change the legend in Fig. 2. If two OTUs belonging to, say, Phyllodocida, are found, rather that using the name “Phyllodocida” for each, I think it is much clearer to use, e.g., Phyllodocida I and Phyllodocida II.
I imagine the authors picked a big polychaete in grab 1, subsample 1… bad luck, but illustrative of what can happen without proper replication!
Line 296. I do not see a “less evident” slope decrease for COI. Certainly not comparing Fig. 5A and C (rather the opposite). May be for the comparison Fig. 5B and 5D. I think this needs to be better formalised. I also suggest adding a third level of accumulation curves: the grab level, with values from 1 to 3 pooled grabs.
Discussion
Guardiola et al 2015 (10.1371/journal.pone.0139633) have compared average similarities within corers, within localities and between localities in deep sediment samples in the W Mediterranean. Likewise, Wangensteen et al 2018 (10.7717/peerj.4705, see Fig. S6) have compared the similarity between ecological replicates, extraction replicates, and PCR replicates in benthic samples (hard substratum). These results can provide useful comparisons here.
I think that the need to replicate grabs within stations is well demonstrated in this study but it is not anything novel, as any ecological study should pick replicate grabs, corers, whatever the sampling device is used. By contrast, the interest in replicating within grabs is new and nicely illustrated by the polychaete problem in subsample1.
While the grab and the within-grab replicates address ecological heterogeneity patterns, the extraction replicates address a technical problem (i.e., whether the amount of sediment used is sufficient). I would separate the two concepts clearly. From the results obtained, using a sample size of 0.5 g five times (five extractions that are pooled afterwards) seems to provide good results. For me, this outcome suggests that the advice should be to use other kits that extract a bigger amount of sediment such as the PowerMax Qiagen or the Norgen soil Maxi Kit. These technical options should be at least mentioned.

Xavier Turon

·

Basic reporting

Hestetun et al conducted an evaluation of spatial replication of North Sea sediment using metabarcoding data of two molecular markers 18S and COI. They analyzed three spatial scales: intra-sample level, grab level and station level using three sampling strategies by collecting eDNA from a single sediment subsample within the same part of a single grab; pooled extraction replicates from different parts of the same grab and pooled extraction replicates from three spatial whole grab replicates from the same station, respectively.
They compare results using common alpha and beta diversity measures such as richness, Shannon diversity and Bray-Curtis pairwise dissimilarity.

They were able to detect a considerable amount of eukaryote richness from a small amount of source sediment using the 18S metabarcoding dataset. In contrast, the COI metabarcoding data showed a high number of prokaryote and unassignable eukaryote reads. They have also produced nice figures evaluating their spatial replications.

However, I found it hard to follow their experiment design and sample labeling which was more clearly described in the Results section only (line 226 to 229). I suggest the authors to improve the description of each sample name inthe M&M section and on figure 1. For example, instead of only showing the label of subsamples I to V and pools A to E in figure 1, the schematic overview of the sampling layout (figure 1, A) could better depict this labeling logic and improve the readability of this labeling nomenclature. This would probably help readers to more easily follow all subsequent analyses as well.

Thus, I would recommend this manuscript for publication and congratulate the authors for such a nice work.

Experimental design

Good experimental design but need to be better described.

Validity of the findings

They were able to detect a considerable amount of eukaryote richness from a small amount of source sediment using the 18S metabarcoding dataset. In contrast, the COI metabarcoding data showed a high number of prokaryote and unassignable eukaryote reads. They have also produced nice figures evaluating their spatial replications.

Additional comments

I would recommend this manuscript for publication and congratulate the authors for such a nice work.

·

Basic reporting

See general comments.

Experimental design

See general comments.

Validity of the findings

See general comments.

Additional comments

This manuscript by Hestetun et al. investigates among/between sample replication and consistency of eukaryote metabarcoding experiments by using a case study of sediment samples from the North Sea amplified with COI and 18S. They show the importance of replication and pooling in mitigating the influence of over-abundant OTUs in single samples. I think the study makes a useful contribution to the field.

While the sample sizes were modest, I thought that the study was well executed, explained clearly, and the data/code were accessible. Overall, I have no serious concerns. The authors may wish to consider some of the specific suggestions/comments below.

Rupert Collins

COMMENTS & SUGGESTIONS

L84: yes, important point.

L178: I can’t see any details of how the reads were demultiplexed. I assume this was done by bcl2fastq on the MiSeq?

L189: the “https://github.com/lanzen/Metamon” link did not work. Looks like it should be "https://github.com/lanzen/Metamon1".

L232: maybe add a mean and standard deviation also?

L387: change “detect an” to “detect a”.

Figures: the colours in some of the figures (especially Fig 2 and Fig 5) are a little garish, and could benefit from more subdued colour palettes.

Figure 3: jittered data points could be superimposed on the boxplots to get a better idea of the spread of the data.

---

## Round 0.2 · accepted · Accept

Dear Jon and co-authors,

I am pleased to accept this manuscript for publication in PeerJ - Congratulations!

I take this opportunity to thank all three reviewers for their time and care in improving the manuscript which will represent a great contribution to the field!

Best,
Xavier

·

Basic reporting

See General Comments

Experimental design

See General Comments

Validity of the findings

See General Comments

Additional comments

The authors have carefully and satisfactorily addressed all my concerns. In my view the ms is ready to go now.

·

Basic reporting

The authors have responded point by point to each concerns I had. Therefore, I would recommend publication as it is.

Experimental design

The authors have responded point by point to each concerns I had. Therefore, I would recommend publication as it is.

Validity of the findings

The authors have responded point by point to each concerns I had. Therefore, I would recommend publication as it is.

Additional comments

The authors have responded point by point to each concerns I had. Therefore, I would recommend publication as it is.